# Complications of the Low Phenylalanine Diet for Patients with Phenylketonuria and the Benefits of Increased Natural Protein

**DOI:** 10.3390/nu14234960

**Published:** 2022-11-23

**Authors:** Nicole McWhorter, Mesaki K. Ndugga-Kabuye, Marja Puurunen, Sharon L. Ernst

**Affiliations:** Synlogic Inc., 301 Binney St. #402, Cambridge, MA 02142, USA

**Keywords:** phenylketonuria (PKU), natural protein intake, medical food, inborn errors of metabolism (IEM)

## Abstract

Phenylketonuria (PKU) is an inherited disorder in which phenylalanine (Phe) is not correctly metabolized leading to an abnormally high plasma Phe concentration that causes profound neurologic damage if left untreated. The mainstay of treatment for PKU has centered around limiting natural protein in the diet while supplementing with medical foods in order to prevent neurologic injury while promoting growth. This review discusses several deleterious effects of the low Phe diet along with benefits that have been reported for patients with increased natural protein intake while maintaining plasma Phe levels within treatment guidelines.

## 1. Introduction

Phenylketonuria (PKU) is a rare inherited metabolic disorder that is characterized by an inability to utilize the essential amino acid phenylalanine (Phe). The worldwide prevalence of PKU is estimated to be 1 per 16,600 newborns, while the prevalence in the United States is estimated to be 1 per 13,500–19,000 newborns [1,2]. PKU is caused by biallelic pathogenic variants in the gene encoding phenylalanine hydroxylase (PAH), which results in partial or complete enzyme deficiency in the liver and accumulation of Phe in the blood and organs. Untreated, the disease results in severe neurological complications, including irreversible intellectual disability [3].

If PKU is detected early and treated with a Phe-restricted diet supplemented with tyrosine (Tyr), outcomes are much improved. In the United States, treatment guidelines identify the range of 120 to 360 μmol/L as the target blood Phe level in appropriately controlled patients irrespective of age [3]. In Europe, guidelines recommend treatment up to the age of 12 years if the Phe blood concentration is above 360 μmol/L, and lifelong treatment is recommended if the concentration is more than 600 μmol/L [4]. This is achieved by severely curtailing protein-containing foods, providing a protein substitute that includes non-Phe amino acids as well as needed macro and micronutrients, and frequent monitoring of blood Phe levels [3,5].

Compliance with a life-long Phe-restricted diet is difficult. Additionally, its health, economic, and quality of life (QOL) impacts continue to be an ongoing challenge for patients and their families. The addition of more natural protein to the diet of PKU patients while still maintaining appropriate metabolic control could serve to ameliorate some of the challenges that are hindering the effective care of these patients. The specific amount of natural protein increase that would be needed in order to achieve these improvements remains unclear. This is complicated by the variability in Phe tolerance among PKU patients by genotype, age, and other factors. However, accruing evidence suggests that patients who can consume more natural protein due to either less severe genotypes or pharmaceutical interventions may have improved outcomes in growth, QOL and risk factors for other chronic diseases, which will be discussed in this review.

## 2. The Challenge of Phe-Restricted Diets

### 2.1. Phenylalanine Tolerance versus Protein Requirements

Phe tolerance varies among PKU patients depending on the amount of residual PAH activity, as well as other factors such as age, growth, and activity level. Most patients with classic PKU tolerate < 500 mg Phe per day (10 g natural protein), while patients with mild to moderate PKU may tolerate < 1000 mg Phe per day (20 g natural protein) [6]. Protein tolerance has been shown to change throughout the life cycle and thus periodic Phe tolerance reassessments should be conducted in order to optimize protein utilization in the body [7].

The low Phe tolerance in the majority of patients results in a large discrepancy between the amount of natural protein patients can tolerate and their overall protein requirements (see Table 1) [5,8]. Protein is a critical macronutrient for the health of all individuals, as it contributes to a number of functions in the body, including the maintenance of muscle mass, bone health, immunity, production of hormones, and satiety. Thus, it is critical that overall protein needs are met in this patient population despite the limited Phe tolerance. This difference must be accounted for by a protein substitute in the form of a medical food in order to supply sufficient nutrients. Medical food usually supplies 75% to 85% of the protein needs of patients with PKU [9].

Recommendations for daily protein intake for children with PKU aged > 4 years is 120% to 140% of the recommended dietary allowance (RDA) based on the American College of Medical Genetics and Genomics (ACMG) 2014 guidelines referenced from the nutritional recommendations developed by the Genetic Metabolic Dietitians International (GMDI), Southeast Regional Newborn Screening, and Genetics Collaborative. The European protein recommendation for children > 10 years is slightly higher than the ACMG guidelines [3,6,10]. These recommendations are based on a factorial method from adult nitrogen balance studies and are not directly determined. More recently, the indicator amino acid oxidation (IAAO) method has been used to determine protein requirements of the general public in several populations including adults, pregnant women, the elderly and healthy children. Using this method Zello et al. found healthy adult men needed 9.1 mg/kg/d of Phe, which would equate to 728 mg Phe (approximately 15 g natural protein) in an adult male with an ideal body weight of 80 kg [11]. This is higher than the lower end of the current recommendations of 290–1200 mg Phe for males aged 19 years and older [12]. In 2017 Turki et al. used this method to quantify the mean protein requirements in 4 children with PKU using a leucine isotope tracer administered orally to directly identify protein requirements. They also found that the actual protein requirements were higher than the current recommendations [13]. This further highlights the large discrepancy between the protein needs of patients with PKU compared to how much natural protein they can tolerate.

**Table 1 nutrients-14-04960-t001:** Comparison of phenylalanine (Phe) tolerance vs. total protein requirements recommended for individuals with phenylketonuria.

Age	Estimated Weight (kg) ^a^	Estimated Phe Tolerance (mg/day) ^b^ [Corresponding Amount of Natural Protein] ^c^	Estimated Total Protein Needs per Day (g) ^d^
0 to <3 months	5.0	130–430 [2.6–8.6 g]	15–17.5
3 to <6 months	7.2	135–400 [2.7–8.0 g]	21.6–25.2
6 to <9 months	8.4	145–370 [2.9–7.4 g]	21–25.2
9 to <12 months	10.2	135–330 [2.7–6.6 g]	25.5–30.6
1 to <4 year	12.2	200–320 [4.0–6.4 g]	≥30
4 to 8 years	21.0	200–400 [4.0–8.0 g]	23.9–27.9
9 to 13 years	36.0	220–500 [4.4–10.0 g]	41.0–47.9
14 to 18 years	61.0	220–1100 [4.4–22.0 g]	62.2–72.6
Adult male	89.8	220–1100 [4.4–22.0 g]	86.2–100.6
Adult female	77.4	220–1100 [4.4–22.0 g]	74.3–86.7

^a^ Ages 0 to 18 years taken from the CDC growth tables for males within the specified age range; 50th percentile determined from https://www.cdc.gov/growthcharts accessed on 16 August 2022 [14]. Mean body weight used for adult male and adult female taken from the CDC publication: Fryar et al. 2018 [15]. ^b^ Acosta 2010 (p. 127, [12]). ^c^ 50 mg Phe averages 1 g natural protein. ^d^ Estimated weight multiplied by estimated protein needs per day. Protein needs per kg weight per day for ages 0 to <4 years taken from Acosta 2010 (p. 69, [12]). For ages 4+ years, 120% to 140% Recommended Dietary Allowance (RDA) for age based on recommendations from Acosta 2010 (p. 127, [12]). RDA by age group taken from the Dietary Reference Intakes (DRI) 2006 [16].

### 2.2. The Complications of the Low-Phe Diet and Protein Substitutes (Medical Food)

The low-Phe diets prescribed for patients with classic PKU are limited to a narrow selection of food groups, including fruits, most vegetables, sugars, pure fats, and medically modified, low-protein food products [17]. Dairy products, meats, legumes, nuts, and most grains contain too much Phe and are typically avoided, which creates challenges both nutritionally and in terms of food palatability and variety. Eliminating these food groups excludes major potential sources of calcium, magnesium, iron, zinc, selenium, vitamin B_12_, and vitamin D, as well as other critical nutrients, such as essential fatty acids and complementary amino acids [5]. The plant-based foods permitted on this low-Phe diet represent incomplete sources of protein, as they fail to provide several essential amino acids, such as lysine [18]. As a consequence, even the protein food sources that patients with classic PKU can eat typically provide neither complete protein to promote satiety nor the complete nutrition necessary for optimal growth and development in children or optimal health in adults [19].

A protein substitute, or medical food, constitutes the main nutritional component in the diet for most patients with PKU. Each protein substitute is comprised of amino acids and also contains fats, carbohydrates, vitamins, and minerals. Historically the amino acid source has been free amino acids, but in recent years there has been the development of medical foods that contain a natural protein low in Phe called glycomacropeptide (GMP). GMP is a 64 amino acid peptide that has been isolated in whey products during cheese production. The medical foods produced with GMP contain 60–70% protein from GMP plus 30–40% supplemental amino acids including the 5 amino acids that are limited in GMP (including histidine, leucine, methionine, tryptophan, and tyrosine) as well as supplemental fats and micronutrients [20,21].

There are several challenges associated with the ingestion of medical foods, especially amino acid medical foods. The first set of challenges is physiological. Because the protein substitute constitutes the majority of nutrients the patient ingests, the diet for these patients is largely synthetic. A synthetic diet is widely understood to be of lesser quality than that derived from whole food sources [22]. Vitamins and minerals in medical foods are not bound to protein and other macronutrients, and thus absorption and utilization are lower than those bound to food sources [22,23]. Synthetic protein substitutes are devoid of phytonutrients and antioxidants, which work synergistically to prevent disease, and are often devoid of prebiotics and fiber as well [24]. Supplements have been shown to be less effective compared to ingestion of nutrients from whole foods, and evidence of this in the PKU population is the presence of micronutrient deficiencies despite adequate supplementation from synthetic medical food [5,25,26]. The protein in amino acid protein substitutes is not absorbed and utilized by the body as well as natural protein, as evidenced by nitrogen balance studies on amino acid protein substitutes [27]. As such, patients on this synthetic diet must have a greater quantity of protein in their diet in order to meet their needs compared to the general population [9,27]. This may lead to deleterious health consequences as well as create a volume issue with patients having to drink large amounts of medical food in order to meet their needs. Even with this large protein volume, because of the rapid oxidation of the free amino acids in the protein substitutes, the synthetic protein cannot suppress ghrelin, a hormone that stimulates appetite, which may cause an increase in food intake or leave the patient feeling hungry [19,27].

The second set of challenges is associated with the poor palatability of the protein substitutes. The protein substitutes must be taken at least three times daily and have been associated with issues including undesirable taste, texture, and smell [27]. Patients have reported great anxiety about taking their protein substitute [28]. Due to these factors, compliance among patients in taking their protein substitute is low [29]. This can lead to protein malnutrition, micronutrient deficiencies, and high blood Phe levels [9,29].

GMP medical foods that are higher in natural protein have helped to ameliorate a few of the challenges associated with medical food intake. Although there have been no kinetics studies comparing GMP vs. amino acid medical foods, Van Calcar et al. did show lower Phe levels after an overnight fast in patients on GMP medical food compared to amino acid medical foods as well as lower blood urea nitrogen (BUN), suggesting a slower release of amino acids and better protein utilization in GMP medical foods [30]. A study by MacLeod et al. found GMP can suppress ghrelin levels more than amino acid mixtures, leading to increased satiety [19]. Reports have noted that GMP products have a better taste profile [19,30]. Despite those advances, GMP medical foods still contain synthetic sources of vitamins and minerals, comprise a large volume that needs to be consumed throughout the day and have been associated with raising Phe levels, especially in those with classical PKU, due to the presence of residual Phe in preparations of GMP [31].

### 2.3. Compliance with Treatment

Adherence to prescribed dietary treatment is fraught with many challenges. Dietary adherence can entail potentially significant costs in time and money for patients and their families, creating a treatment barrier for many [28,32,33,34]. In addition to cost barriers, the diet itself is difficult to follow, as patients have found the diet and protein substitutes to be unpalatable, socially isolating, and overly restrictive [35]. In a large European study, compliance with treatment decreased from 70% in childhood to 20% after the age of 15 [29]. The study authors state that the diet is difficult to follow and suggest that recommended blood Phe levels may be unachievable for most patients with dietary management. Thus, they state new treatment modalities must be developed in order to prevent the detrimental effects of high Phe levels [29]. A survey conducted by the National PKU Alliance (NPKUA) in 2015 found that less than half of the 625 respondents were able to achieve Phe levels within the treatment range despite more than two-thirds having the desire to do so, with 51.7% of respondents stating that following their treatment was difficult and 91.4% stating that the development of new treatments for PKU was important to them [36]. When dietary non-adherence is present in children, it can elevate blood Phe levels and have severe detrimental effects on long-term neurocognitive outcomes [37,38] Even in adulthood, dietary non-adherence with resulting high blood Phe levels is not inconsequential. Suboptimal nutritional outcomes and adverse neuropsychological outcomes are observed in non-adherent adult patients [39]. Many patients stop adhering to strict dietary control as adolescents or adults but often continue to restrict high-protein natural foods without appropriate protein substitute supplementation. This can put them at risk for the development of overt nutritional deficiencies in addition to complications of high blood Phe levels [5].

## 3. Adverse Effects on Growth, Health, and Nutritional Outcomes Resulting from the Low-Phenylalanine Diet

The effectiveness of the low-Phe diet (supplemented with appropriate medical food) in the prevention of gross neurologic impairments caused by high blood Phe levels in patients with PKU is widely acknowledged. Early detection and effective treatment can result in patients achieving normal intelligence quotients (IQs) and living productive lives [38]. However, a diet so low in natural protein can have adverse impacts on growth, health, and nutrition.

### 3.1. Growth and Body Composition

Optimal growth and development have been a concern for patients with PKU since the introduction of the semi-synthetic diet in 1951 by Bickel and associates, with early studies indicating poor growth in patients with PKU compared to controls [40]. Historic studies found that normal linear growth may be attainable in some children with PKU but requires a protein equivalent intake greater than the recommended Dietary Reference Intake (DRI), thus leading to an increased risk for obesity [41]. More recent studies, however, have shown conflicting outcomes regarding growth, with several finding optimal growth outcomes are often not obtained in children with severe PKU [8,42,43]. Children following a low-Phe diet had lower linear growth and weight for age than the reference populations, while children with mild hyperphenylalaninemia (MHP) who were not on restricted diets did not. This difference was particularly notable within the first 3 years of life when nutrition has the greatest impact on growth and development [8]. Although some studies have indicated more normal growth in children with PKU, on closer examination, this trend, when observed, was likely a consequence of a relaxed diet indicating that better growth outcomes were associated with poor adherence to the low-Phe diet and an increase in natural protein intake [8]. Several studies have reported smaller head circumferences for children with PKU compared to controls [44,45,46,47]. A large Dutch study conducted on 174 infants with PKU found a positive, statistically significant relationship between head circumference and natural protein intake as well as total protein intake, but no relationship with protein substitute intake or total calories [46,47].

Studies on body composition of patients with PKU have been conflicting with no meta-analysis done yet to date. Several studies have shown no statistically significant differences in body composition of patients with PKU compared to controls; however other studies have found a higher percentage of body fat and a lower percentage of lean body mass in patients with PKU compared to the general population, with higher BMI being associated with lower natural protein intake [37,47,48]. Furthermore, a lower natural protein intake has been associated with higher BMI, waist circumference, higher prevalence of obesity and metabolic syndrome, and insulin resistance [49,50]. Total macronutrient imbalance may explain some of these findings. Children with classic PKU eat 50% to 100% more high-energy, sugar-containing drinks, sweets, chips, and cookies than their peers, as these foods are lower in natural protein. Additionally, these children also experience higher rates of food neophobia than controls, which also negatively contributes to body composition issues [32].

### 3.2. Bone Health

Higher rates of osteoporosis and bone fractures have been noted in patients with PKU [51,52]. According to the 2016 National Institutes of Health (NIH) Treatment Consensus, abnormalities in bone metabolism in patients with PKU may not be directly correlated with vitamin D or calcium intake, and the exact etiology remains unknown [9]. Animal models have demonstrated that high Phe levels and diets that include protein substitutes are correlated with worse outcomes for bone health [52,53]. A study of mice receiving a GMP diet (with more natural protein) versus those fed an amino acid diet indicated better bone outcomes on the GMP diet [51]. Studies in PKU patients have indicated a correlation between poor bone outcomes and low natural protein intake [54]. Stroup et al. found increased urinary calcium and magnesium excretion as well as a higher potential renal acid load in patients on amino acid medical food compared to those on GMP medical foods and hypothesized a negative correlation between bone health and the renal acid load from the amino acid medical foods [55]. Patients with mild PKU and those treated with sapropterin dihydrochloride (with increased natural protein intake) have shown better bone health outcomes [54].

### 3.3. Micronutrient Deficiencies

Due to the limited intake of natural protein, micronutrients need to be supplemented in protein substitutes to prevent overt nutritional deficiencies. However, despite large volume supplementation, maintaining sufficient vitamin and mineral levels continues to be a challenge in patients with PKU. Protein substitutes contain vitamins and minerals according to national guidelines for the required amount of micronutrients by age. These recommendations do not account for the reduced bioavailability or lack of nutrient interactions that result from excluding entire food groups from the diet. Serum levels of some micronutrients remain low despite adequate intake, indicating limited bioavailability [48]. Deficiencies in selenium, zinc, and iron have been reported in patients with PKU [5,25,26]. Patients who discontinue or reduce the intake of their protein substitute without a commensurate increase in their natural protein intake are at risk for overt micronutrient deficiencies [5]. It has been noted that patients taking sapropterin dihydrochloride on a more liberalized diet have a reduced incidence of low serum levels of some micronutrients compared to those on a lower protein diet, despite having less intake of said nutrients [56].

### 3.4. Fatty Acids

The low Phe diet is typically low in natural protein from sources that are also naturally high in fats, for example, salmon, walnuts, and almonds. This may result in lower intakes of long-chain polyunsaturated fatty acids and docosahexaenoic acid (DHA), which can further compromise neurodevelopment. Studies have demonstrated improvements in motor functioning when DHA has been supplied to patients [57,58,59,60].

### 3.5. Other Health Concerns

In an insurance claim-based study assessing comorbidities among 3691 patients with PKU, the authors found an increased rate of comorbidities, such as asthma, alopecia, urticaria, gallbladder disease, rhinitis, esophageal disorders, anemia, overweight, gastroesophageal reflux disease, eczema, renal insufficiency, osteoporosis, gastritis/esophagitis and kidney stones in patients with PKU compared to matched controls [61]. The highest adjusted prevalence ratio (PR) was found for renal insufficiency with hypertension (PR [95% confidence interval (CI)]: 2.20 [1.60 to 3.00]). Concerns regarding an increased risk of kidney disease in PKU patients have also been noted in other studies [62]. The high amino acid load from the protein substitutes is thought to explain the increased risk of kidney disease. The next highest adjusted prevalence ratio was found for overweight and obesity (PR [95% CI]: 2.06 [1.85 to 2.30]) [61].

## 4. Benefits of Adding Natural Protein to the Diet

Several studies have led to an accumulation of data on the outcomes of increased intake of natural protein in patients with PKU, including the effects on Phe tolerance, blood Phe levels, growth and development, quality of life, compliance, and costs. It is currently unclear what amount of additional natural protein intake makes a clinical difference in patients with PKU. However, by looking at studies that compare patients on a Phe-restricted diet compared to those consuming more natural protein due to increased Phe tolerance naturally or via pharmacological intervention, clear benefits to increased natural protein intake can be seen.

### 4.1. Growth, Development & Micronutrients

Growth, head circumference, and body composition have all been positively correlated with increased natural protein intake. Children with MHP and children with higher natural protein intake due to adjunct therapy with sapropterin dihydrochloride have growth rates that are more similar to those observed in the general population than those observed in patients with classic PKU on more limited natural protein intake [8]. Children with poor dietary compliance have more normal growth outcomes as well [8]. A positive correlation between increased natural protein intake and head circumference and a negative correlation with body fat percentage have been reported [8]. These same effects have been observed in children with higher natural protein intake due to sapropterin dihydrochloride [48]. Children using GMP medical foods with more natural protein have been shown to have a positive trend of growth and lean body mass compared to children on amino acid medical foods [31]. Bone health and micronutrient serum levels are also positively correlated with increased natural protein intake [54,56].

### 4.2. Quality of Life (QOL)

Increased natural protein intake has been associated with improved QOL for patients with PKU. Being able to eat more natural protein, and thus more “normal” foods, accompanied by a decrease in the prescribed amount of protein substitutes contribute to improved QOL. A multicenter European study assessing the critical factors impacting QOL in PKU patients found that anxiety regarding blood Phe levels, guilt related to poor dietary adherence, and poor palatability of the protein substitute all negatively impacted QOL [28]. The QOL impact factor of using a protein substitute was highest in those with classic PKU and strict natural protein restriction compared to those on a more relaxed diet. Those with more dietary restrictions also reported higher dietary non-adherence compared to those with mild or moderate PKU [28]. The QOL of patients treated with sapropterin dihydrochloride on a more liberalized diet was less impacted by the condition and adherence to the protein substitute was higher. In a long-term study of patients taking sapropterin dihydrochloride, an improvement in QOL metrics was reported by 50% of patients (14% reported no change and no information was available for 36%) [63]. Concerning the quality of life, the NIH Consensus Statement reads:

“Two categories of individuals with PKU who are BH4 responsive may realize improved quality of life and nutritional benefits. First, in some patients who are unable to adhere to dietary therapy or unable to maintain a level of dietary restriction and medical food intake that sufficiently controls blood Phe, sapropterin may lower blood Phe to an acceptable range without further dietary modification. Second, patients whose dietary therapy and adherence already maintain blood Phe levels within a therapeutic range may be able to increase intake of natural protein, positively impacting nutritional status. Quality of life is further enhanced with the ability to eat more normal sources of protein” [38].

In the 2015 survey by the NPKUA, patients reported that increasing natural protein intake (77.7%), being able to eat any foods they would like regardless of protein content (76%), and decreasing consumption of medical food (57.7%) would have the highest impact on QOL [36].

### 4.3. Compliance

Increased natural protein intake improves overall metabolic control and diet adherence. In a large 18-year longitudinal study in British Columbia, the authors noted that blood Phe concentrations were more often within the desired therapeutic range in those with a higher Phe tolerance [64]. One study noted an increase in compliance to treatment in 63% and an improvement in dietary compliance in 47% of patients with increased natural protein tolerance due to sapropterin dihydrochloride use [63]. In the 2015 NPKUA survey, the respondents on sapropterin dihydrochloride treatment had lower Phe levels than the other respondents and were more likely to consider their treatment “easy to manage” [36].

### 4.4. Financial Cost and Time Burden

Dietary treatment for PKU presents a challenge financially and time-wise. Out-of-pocket costs for families may include the protein substitute, specialized low-protein food products, gram scales for weighing food and medical food, medical office visits, Phe monitoring laboratory measurements, postage for mailing Phe blood tests to the laboratory, extra luggage expenses for traveling with medical food, and attendance at PKU events (camps, low-protein cooking classes, etc.) [34]. In some states/countries, the protein substitute may be covered by health insurance; however, this is not the case in most areas of the United States. Additionally, other aspects of care, including specialty low-protein food products, are often not covered by insurance. For patients with moderate/classic PKU, the high cost of the low-protein foods and protein substitutes accounts for non-compliance issues in 20% to 30% of families [33]. The burden is significantly lower for patients with mild PKU who can consume more natural sources of food and do not have to rely on expensive medically modified foods for the majority of their diet.

Time demands are also significant for patients and caregivers, as medical management and preparing specialty foods can be time-consuming. As packaged foods available for PKU are limited and expensive, almost all food must be homemade, including common items such as bread. Commonly available convenience foods, including those available at most restaurants, often contain too much Phe and therefore cannot be used to save time on meal preparation. Most families must make multiple meals a day- those for the patient with PKU and those for the rest of the family, as a low-Phe diet would be nutritionally inadequate for those without PKU. A study in the Netherlands indicated that severe PKU (protein tolerance ≤ 10 g) was associated with a significantly greater time burden (median 595 h per year [interquartile range 214 to 862]) than mild PKU (median 235 h per year [interquartile range 123 to 353]), and out-of-pocket costs were significantly higher for patients with severe PKU than mild PKU [34]. This time is significant, as it could be used in other income-generating ways. The study authors estimated the cost of lost time (in 2013) would equate to EUR 7066 (USD 7920) for caregivers and EUR 2341 (USD 2625) for patients annually (based on a common labor wage) [34].

## 5. Pharmacotherapies Leading to Increased Natural Protein

Several adjunct therapies have been developed which have helped patients increase their natural protein intake. Unfortunately, the currently approved therapies are not effective for all patients with PKU and thus leave the majority of patients lacking options to help increase natural protein intake. Additionally, not all approved or emerging therapies allow for unrestricted natural protein intake. With that in mind, patient education on food choices becomes imperative in order to maximize the benefits of the natural protein they can consume.

### 5.1. Natural Protein Increases with Adjunct Therapies

Sapropterin dihydrochloride is an oral medication approved by the FDA in 2007 that is a synthetic form of tetrahydrobiopterin (BH4), the co-factor for the PAH enzyme. Sapropterin works by stabilizing the PAH enzyme, thus keeping it more active in some patients with residual PAH enzyme activity [65,66]. A large metanalysis found a 2- to 4-fold increase in Phe tolerance with sapropterin treatment [67]. The mean Phe tolerance in pediatric patients taking sapropterin has been shown to increase from 18 mg/kg before treatment to 40 mg/kg during treatment [68]. Sapropterin is only effective for those with mild/moderate PKU, which is approximately 30% of patients [69].

Pegvaliase is an injectable enzyme substitution therapy approved for adults in 2018 that converts Phe to trans-cinnamic acid and ammonia [70]. After 24 months on pegvaliase, patients in clinical trials had a 68.7% decrease in blood Phe levels and were able to increase natural protein intake and decrease their protein substitute [71]. Extensive long-term studies on pegvaliase that demonstrate an increase in Phe tolerance in PKU patients are pending, and prior clinical studies did not report total increases in natural protein. However, a few smaller studies have reported natural protein intake in patients using pegvaliase while maintaining Phe levels < 360 μmol/L. One study of 18 patients showed a natural protein intake of 73.2 ± 17.6 g protein/day (1.0 ± 0.3 g/kg/day) on pegvaliase [72]. Another study comparing six adult patients on pegvaliase to six patients on diet therapy showed a natural protein intake of 78.0 ± 24.9 g/day (1.0 ± 0.4 g/kg) on pegvaliase versus 24.6 ± 5.0 g/day (0.3 ± 0.05 g/kg) on diet therapy [73]. It is generally acknowledged that patients on pegvaliase will consume a liberalized diet without a protein substitute [74]. However, despite its effectiveness in lowering blood Phe levels, this therapy has been associated with many adverse reactions, including high rates of hypersensitivity reactions and anaphylaxis [75]. In the 2015 NPKUA survey, patients indicated a desire for additional natural protein in their diet but preferred an adjunct oral medication to a daily injection, such as pegvaliase [36].

Several other emerging therapies under investigation may be able to help patients with natural protein intake. Homology Medicines, Inc is currently conducting a Phase 1/2 study to evaluate the efficacy of its gene therapy [76]. PTC Therapeutics has developed an oral form of sepiapterin, a precursor to intracellular BH4, and is currently in Phase 3 of the development process [77,78]. Synlogic Inc. is developing a synthetic biotic medicine (a probiotic engineered to consume Phe) and has completed a Phase 2 clinical trial [79,80]. The amount of natural protein intake on these emerging therapies is still unknown, however, they present the potential to expand the number of patients with ability to increase natural protein intake and benefit from the reported improvements associated with this increase.

### 5.2. Maximizing Benefits of Natural Protein Increase

To maximize the benefits of an increase in natural protein intake, it is critical that the natural protein added is of high quality. To ensure optimal utilization of protein, it is important that the protein contains all amino acids needed for protein anabolism. Animal products or a combination of plant products such as grains plus legumes can serve as sources of high-quality protein. Studies have shown that patients often choose foods with lower-quality protein such as increased portions of potatoes and pasta [68,81,82]. Therefore, diet counseling and education are critical to ensure the maximal clinical benefits of an increase in natural protein intake.

An additional benefit to increasing natural protein intake is the opportunity to decrease the volume of protein substitute the patient requires. Recommendations on increasing natural protein intake along with pharmacological interventions encourages a proportional decrease in protein substitute [67]. However, as the protein substitute is decreased, careful attention must be paid to the quantity and quality of the increased natural protein intake to prevent protein and micronutrient insufficiency. Some studies of patients on sapropterin who have decreased their protein substitute have shown a marked decrease in micronutrient intake because their food choices did not include replacement items with sufficient micronutrients [67,83].

## 6. Future Directions

Additional research is needed to elucidate further benefits of increased natural protein intake for patients with PKU. Outcome studies that are stratified by actual natural protein intake and controlled for compliance in medical food dosing would help to reduce the potential conflicting variables that are present in many of the current studies. Nitrogen balance studies comparing patients with various natural protein prescriptions could help us understand better the minimum natural protein intake needed for ideal protein synthesis. As new therapies are emerging and patients are able to increase their natural protein intake, this is an ideal time for clinicians to develop a systematic evaluation process to measure changes in growth, compliance, micronutrient status, QOL, and overall diet quality. These measures can be vital in evaluating the long-term impact of dietary changes.

## 7. Summary

The available literature indicates significant benefits to increasing natural protein intake in the diets of patients with PKU. Better outcomes in growth, metabolic health, nutritional status, and QOL are correlated with increased natural protein intake. Currently, therapeutic options remain limited, and additional effective therapies are needed for patients with PKU. Therapies aimed at increasing natural protein intake are especially needed for children, in whom optimal growth and development are contingent upon appropriate nutrition. In adolescents and adults, increased natural protein intake can improve dietary compliance and lead to improved neurocognitive function. The PKU community would benefit from additional therapeutic options that would allow patients to increase natural protein intake as well as continued research focusing on the benefits of natural protein increases.

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
