# Peer review of "Complications of the Low Phenylalanine Diet for Patients with Phenylketonuria and the Benefits of Increased Natural Protein"

_nutrients, 2022, doi:10.3390/nu14234960_

Round 1
Reviewer 1 Report
Well-written comprehensive review of natural protein intake in PKU. Minor edits suggested:
1. The manuscript would benefit from a future directions section. What studies would be needed to demonstrate benefits to increased natural protein? The authors mention no AA kinetics with GMP. What about protein synthesis studies in patients consuming normal natural protein vs medical foods?
2. The introduction ends with a sentence indicating improved outcomes with more natural protein. Please indicate what outcomes here. Growth, micronutrient intake, quality of life. I think it important to make it clear that the outcomes you are focusing on are not neurocognitive d/t the relationship to high natural protein and high Phe in the absence of other treatments.
Author Response
Dear Reviewer,
Thank you so much for your time to review our manuscript. Per your recommendations, we have made the following changes:
Point 1. The manuscript would benefit from a future directions section. What studies would be needed to demonstrate benefits to increased natural protein? The authors mention no AA kinetics with GMP. What about protein synthesis studies in patients consuming normal natural protein vs medical foods?
Response 1. A future directions section was added that suggested future nitrogen balance studies as well as more well controlled studies. It reads "Additional research is needed to elucidate further benefits of increased natural protein intake for patients with PKU. Outcome studies that are stratified by actual natural protein intake and controlled for compliance in medical food dosing would help to reduce the potential conflicting variables that are present in many of the current studies. Nitrogen balance studies comparing patients with various natural protein prescriptions could help us understand better the minimum natural protein intake needed for more ideal protein synthesis. As new therapies are emerging and patients are able to increase their natural protein intake, this is an ideal time for clinicians to develop a systematic evaluation process to measure changes in growth, compliance, micronutrient status, QOL, and overall diet quality. These measures can be key to evaluating the long-term impact of dietary changes." See page 10 lines 451-462.
Point 2. The introduction ends with a sentence indicating improved outcomes with more natural protein. Please indicate what outcomes here. Growth, micronutrient intake, quality of life. I think it important to make it clear that the outcomes you are focusing on are not neurocognitive d/t the relationship to high natural protein and high Phe in the absence of other treatments.
Response 2. The introduction was edited per your recommendation. It now reads "However, accruing evidence suggests that patients who can consume more natural protein due to either less severe genotypes or pharmaceutical interventions may have improved outcomes in growth, QOL and risk factors for other chronic diseases, which will be discussed in this review." See page 2 lines 48-51.
Thank you again for your review.
Sincerely,
Nicole McWhorter, MS RD
Reviewer 2 Report
This article is supportted by a company, is a very interesting review of the literatura in relation to this topic.
I Think there must be a mistake in the table 1 Pg 72:
14-18 years, adult male and adult female in the 3ª columne, estimated phe tolerance (mg/day) 220-110 it must be 220-1100
The bibliographic review i very adecuate and extensive
Author Response
Dear Reviewer 2,
Thank you for your time in reviewing our manuscript. We have made corresponding corrections to the paper.
Point 1: I Think there must be a mistake in the table 1 Pg 72:14-18 years, adult male and adult female in the 3ª columne, estimated phe tolerance (mg/day) 220-110 it must be 220-1100
Response 1: Thank you for seeing this error. The number has been corrected to 1100. Please see Table 1 on page 2 line 71.
Again we appreciate your review.
Thank you,
Nicole McWhorter, MS RD